# Canadian Guideline on the Management of a Positive Human Papillomavirus Test and Guidance for Specific Populations

Tiffany Zigras [1,*], Marie-Hélène Mayrand [2], Celine Bouchard [3], Shannon Salvador [4], Lua Eiriksson [5], Chelsea Almadin [6], Sarah Kean [7], Erin Dean [7], Unjali Malhotra [8], Nicole Todd [9], Daniel Fontaine [10] and James Bentley [11]

1 Trillium Health Partners, Department of Obstetrics and Gynecology, University of Toronto, Toronto, ON M5G 1E2, Canada
2 Département d'obstétrique-gynécologie, Université de Montréal, Montreal, QC H3C 3J7, Canada
3 Clinique de Researche en Sante des femmes, Quebec City, QC G1V 3M7, Canada
4 Department of Obstetrics and Gynecology, McGill University, Montreal, QC H3A 0G4, Canada
5 Department of Obstetrics and Gynecology, McMaster University, Hamilton, ON L8S 4L8, Canada
6 Health Innovation and Evaluation Hub, University of Montreal Hospital Research Centre, Montreal, QC H3Z 2H5, Canada
7 Department of Obstetrics, Gynecology and Reproductive Sciences, Winnipeg, MB R3J 3T7, Canada
8 Office of the Chief Medical Officer, First Nations Health Authority, West Vancouver, BC V7T 1A2, Canada
9 Department of Obstetrics and Gynecology UBC, Vancouver, BC V5Z 4E1, Canada
10 Department of Pathology and Laboratory Medicine, Valley Regional Hospital, Kentville, NS B4N 5E3, Canada
11 Department of Obstetrics and Gynecology, Dalhousie University, Halifax, NS B3H 4R2, Canada
* Correspondence: tiffany.zigras@thp.ca

**Abstract:** The purpose of this paper is to provide evidence-based guidance on the management of a positive human papilloma virus (HPV) test and to provide guidance around screening and HPV testing for specific patient populations. The guideline was developed by a working group in collaboration with the Gynecologic Oncology Society of Canada (GOC), Society of Colposcopists of Canada (SCC), and the Canadian Partnership Against Cancer. The literature informing these guidelines was obtained through a systematic review of relevant literature by a multi-step search process led by an information specialist. The literature was reviewed up to July 2021 with manual searches of relevant national guidelines and more recent publications. The quality of the evidence and strength of recommendations were developed using the Grading of Recommendations Assessment, Development, and Evaluation (GRADE) framework. The intended users of this guideline include primary care providers, gynecologists, colposcopists, screening programs, and healthcare facilities. The implementation of the recommendations will ensure an optimum implementation of HPV testing with a focus on the management of positive results. Recommendations for appropriate care for underserved and marginalized groups are made.

**Keywords:** human papilloma virus test; HPV test; HPV self-sampling; HPV guidelines; Indigenous; First Nations; Metis; LGBTQ2S+; immunocompromised; newcomers; immigrants

## 1. Introduction

In 2020, the World Health Organization (WHO) announced a global strategy towards the elimination of cervical cancer as a public health priority by reducing the incidence of cervical cancer to less than 4 per 100,000 [1]. Most cervical cancer is human papilloma virus (HPV)-related and preventable. Over the past 40 years, secondary prevention with screening and treatment of pre-invasive lesions has contributed to significantly reduced incidence and mortality. Primary prevention with the HPV vaccination is expected to further reduce the incidence. Despite this, we still see approximately 1450 new cases and 380 deaths from cervical cancer per year in Canada [2]. In line with the WHO recommendations, the Canadian Partnership Against Cancer has proposed an action plan for the elimination of

cervical cancer in Canada by 2030 [3]. The priorities of this plan include the implementation of cervical cancer screening that primarily uses HPV testing and improved follow-ups of abnormal screening results. Primary HPV testing currently has not been implemented. However, it is anticipated that this will happen soon. The age of initiation, cessation, and interval are controlled at the provincial level and were beyond the scope of this document.

The introduction of cytologic screening programs has decreased the incidence of cervical cancer. However, cytologic screening programs have failed to eliminate cervical cancer due to issues that include test limitation and sub-optimal coverage of screening activities. Randomized controlled trials (RCTs) have demonstrated that cervical cancer screening with HPV testing offers higher protection against cervical precancer and cancer compared to cytology-based screening [4–8]. In 2021, the International Agency for Research on Cancer (IARC) reported that, overall, HPV DNA testing is more sensitive than cytologic analysis for the detection of CIN2+ and is associated with reduced detection rates of CIN2+ in subsequent screening rounds, as well as a greater reduction in the incidence of cervical cancer than cytologic analysis when the same screening interval is used [9]. According to a recent report, 48 (35%) of 139 countries have recommended HPV-based cervical screening. However, most are currently transitioning from cytology to HPV testing [10].

In this context, a guideline was commissioned by the Canadian Partnership Against Canada in collaboration with the Gynecologic Oncology Society of Canada (GOC) and the Society of Canadian Colposcopists (SCC) to provide an evidence-based approach to the management of a positive HPV test. Guidance was also sought for caring for specific populations, such as those who are immunocompromised and those who face challenges in accessing and receiving quality care after a positive screening test result, such as LGBTQ2S+, First Nations, Inuit, Metis, newcomers, and immigrants to Canada. We were also asked to review the evidence surrounding HPV self-sampling in the general population and in non-/under-attenders and regular attenders.

The group does not endorse any one HPV test over another. Any Health Canada-approved HPV test or device for cervical screening should be used for testing according to their regulatory approval. Recommendations provided in this document are not meant to supersede local guidelines or existing provincial or territory population-based cervical screening programs or clinical judgment.

## 2. Methods

The committee was recruited to represent interested gynecologists and pathologists from many Canadian provinces. Initial meetings were held in the fall of 2020 and a series of objectives was developed.

The literature informing these guidelines was located using a multi-step search process led by an information specialist. An initial search for existing clinical practice guidelines related to HPV testing for cervical screening was designed in Ovid MEDLINE All and performed on 7 June 2021. A modified version of the guidelines filter developed by Canadian Agency for Drugs and Technology in Health (CADTH) for Ovid MEDLINE was used [11]. The search is reported in full in Supplementary Table S1.

A second search targeting evidence related to HPV testing for cervical screening in specific contexts and populations was designed in Ovid MEDLINE All and executed on 10 November 2021. The contexts of interest included triaging a positive HPV test, self-sampling in the general population, and self-sampling in non-attenders. Specific searches were also performed to summarize the existing literature on HPV testing in populations of interest, including the Indigenous, LGBTQ+, immunocompromised, immigrants, new-comers to Canada, and those living in rural settings. A modified version of the LGBT search filter from Lee et al. was used to develop the LGBTQ+ portion of the search [12]. A publication date limit from 2018–present was applied to capture the recent evidence. No publication type, study design, or language limits were applied. The search is reported in full in Supplementary Table S2.

A third search targeting evidence related to managing a positive HPV test after a hysterectomy was designed in Ovid MEDLINE All and executed on 7 February 2022. A publication date limit from 2018–present was applied to the search. No publication type, study design, or language limits were applied. The search is reported in full in Supplementary Table S3.

For each specific theme searched, the results were imported to Covidence for deduplication and screening. Members of the working group screened titles and abstracts for relevant papers, which were then obtained for full text review.

Our working group set cut off risk of ≥5% of CIN3+ to be referred to colposcopy and used this to guide referral algorithms to colposcopy after a positive HPV test. During the working group meetings, the committee was asked to generate and propose algorithms for self-sampling and referral to colposcopy, understanding that there is still no FDA or Health Canada approved test for self-sampling. This is still under evaluation in Canada.

Recommendations were developed and graded based on the quality of evidence available using the Grading of Recommendations, Assessment, Development and Evaluations (GRADE) framework Supplementary Table S4. Where evidence was limited, expert consensus recommendations were generated by discussion of the guideline committee members. These recommendations, along with summaries of the supporting evidence, are presented here.

## 3. Results

### 3.1. HPV Positive Test Management

Recommendations:

- HPV tests providing partial genotyping information are preferred. (strong, high)
- Reflex liquid-based cytology should be performed on all HPV-positive samples. (strong, high)
- Persons testing positive for HPV 16 or 18 should be referred for colposcopy, regardless of reflex testing result. (strong, high)
- Persons testing positive for other HR types should be referred to colposcopy if their cytology shows AGC, ASC-H, HSIL, or cancer. (strong, high)
- Persons testing positive for other high types with normal, ASCUS, or LSIL cytology should not be referred to colposcopy but retested (HPV and reflex cytology) at 12 and 24 months. If, at 24 months, there is persistent HR–HPV positivity, regardless of cytology, they should be referred to colposcopy. (strong, high)
- There is currently insufficient evidence to support the use of p16 and DNA methylation as triage tools following a positive HPV test. (conditional, low)

The Pap test has been an effective screening strategy. It is responsible for reductions of up to 70% of cervical cancers where quality screening programs have been implemented. However, the low sensitivity and interpretive nature of the test are significant limitations. When compared in the Canadian context, Mayrand et al. showed that the sensitivity of HPV testing and Pap testing to detect CIN2+ was 94.6% and 55.4%, respectively, with a specificity for HPV of 94.1% and 96.8% for Pap testing [13]. Abnormal Pap cytology results, such as ASC-H, HSIL, and AGC, are classified as having a higher risk for CIN2+ and are directed to colposcopy, whereas ASCUS and LSIL are equivocal. Current triage strategies for ASCUS and LSIL used in Canada consist of either repeat cytology or HPV testing, with cytology showing an ASCUS result where only 50% of cases are HR–HPV related; many cases can be triaged back to regular screening [14]. Several large-scale randomized control trials in multiple populations have conclusively shown primary HPV testing to be superior to Pap testing in detection of precancerous changes i.e., CIN2+ [4–7,15–17]. Ronco evaluated four European trials in 2014 that had shown efficacy in precursor detection, while metanalyses showed a 0.6 (95% CI 0.40–0.89) reduction in invasive cervical cancer rates [8].

The evidence from these trials has shown that HPV testing alone would be more sensitive but also would cause many individuals to be referred to colposcopy unnecessarily.

Hence, there is a need for an appropriate triage test. The ideal triage test post-HPV testing would allow for full automation and avoid tests that have inherent subjectivity.

We are proposing a management strategy following a positive HPV test after evaluating the existing literature on triage tools to best identify CIN2+ using a general cut-off of >5% risk of CIN3+. There is no universally accepted level of risk established [18]. It is recommended that, as HPV testing is implemented into cervical cancer screening programs, cervical screening programs monitor their performance and use their local data to inform their own practice guidelines. The recommendations provided here are based on the current available evidence. However, it should be noted that the risk of CIN3+ may be higher in patients who have not participated in routine screening and who have a history of abnormal cervical screening test results [19]. It is anticipated that in all populations, the CIN3+ risk will likely decrease in subsequent rounds of screening and proposed algorithms may need to be adjusted.

### 3.1.1. Triage by Cytology

Triage by cytology utilizes the high sensitivity of HPV testing combined with the specificity of cytology. It also has the advantage of being performed on the same sample as the HPV test if liquid-based cytology (LBC) is conducted. The HPV Focal trial performed in BC evaluated a protocol with HPV testing followed by reflex LBC testing for those who tested positive for HPV compared to liquid-based cytology alone. If the LBC was abnormal (≥ASCUS), a referral was made to colposcopy. The control arm was managed similarly to the standard of care using reflex HPV testing and conventional criteria for referrals to colposcopy. Testing was done at 0, 24, and 48 months. The use of HPV primary testing resulted in significantly fewer CIN3+ and CIN2+ cases at 48 months. For CIN3+, the risk reduction was 0.42 (95% CI 0.25–0.69). For CIN2+, it was 0.47 (95% CI 0.34–0.67). However, this benefit came with a marked increase in referrals to colposcopy in the first round of screening in the HPV arm. By the 48-month mark, there was less colposcopy in this arm, with the overall rate being similar [15]. In the intervention arm of the Swedescreen trial, they analyzed 11 different strategies for detecting CIN3+ and compared this to cytology alone. HPV testing, followed by cytological triage and a repeat HPV testing of cytological normal HPV positive results, increased the sensitivity by 30% to 54% and did not significantly increase the number of tests performed [6]. This approach has increased referral rates to colposcopy, particularly initially as has been reported in the initial evaluation of the Australian implementation, where colposcopy rates increased from 3.5% to an estimated 6.2% in the first two years and even up to 12% in the youngest age group who are at the lowest risk of significant disease [20].

### 3.1.2. Triage by HPV Genotype

Persistent HPV infection is a necessary cause of cervical cancer. There are more than 100 known types of HPV. Currently, 13 high-risk genotypes have been identified as carcinogenic [21]. Among these HR–HPV genotypes, HPV 16 and HPV 18 are responsible for 70% of cervical cancers [22,23]. We also know that a majority of HPV infections will clear on their own, especially in individuals < 40 years old [24–26]. Long-term infections with HR–HPV can lead to cervical cancer if precursor lesions are left undetected and untreated. The natural history would suggest that this usually occurs over 10–15 years from an initial infection of HR–HPV [27–30]. Studies have shown that of the HPV tests collected, 90% of patients tested will be HR–HPV negative, 3% will have HR–HPV 16/HR–HPV 18 positive, and 7% will have HR–HPV other (Table 1) [31]. Recently introduced HPV tests are able to differentiate HPV types that will allow for the stratification of risk. Data has shown that the risk of CIN3+ is between 10.6% and 25% for those who are HPV 16 positive, and 5.89 to 11% for HPV 18 positive regardless of cytology results [32,33]. Newer tests are available that allow even further stratification of HPV genotypes by providing information about other HR–HPV types such as HPV 31, 45, etc. The high rate of CIN3+ following an HPV test showing HPV 16 or 18 positive allows for direct referral to colposcopy, regardless of the

cytology result. This approach has been adopted by many jurisdictions, including Australia and USA. HPV "other" results, with negative HPV 16 and HPV 18, do not have as high a risk of CIN3+. Thus, they do not meet the threshold for direct referrals to colposcopy without further triage [20,34].

**Table 1.** Immediate Risk of HSIL(CIN3+) by HPV and cytology.

| | HPV | | | |
|---|---|---|---|---|
| **Cytology** | **Pos HR HPV (Any)** | **Pos HPV 16** | **Pos HPV 18** | **Pos HPV Other** |
| Normal | 3.4% [31] | 5.3% [31] | 3% [35] | 2% [35] |
| ASCUS | 4.4% [34] | 9–12.9% [31,36] | 5% [36] | 2.7–4.4% [34,36] |
| LSIL | 4.3% [34] | 11% [31] | 3% [35] | 4.3% [34] |
| ASC–H | 26% [34,35] | 28% [31,35] | 15% [31] | 26% [34,35] |
| HSIL | 49% [34,35] | 60% [31,35] | 30% [31,35] | 49% [34,35] |

### 3.1.3. Triage by Genotyping and Cytology

This approach has been recommended by the WHO where genotyping information is available and by many national screening programs [1]. This allows for minimal extra testing with appropriate triage to and away from colposcopy. Individuals with a HR–HPV 16 positive and/or HR–HPV 18 positive test will go directly to colposcopy, whereas HR–HPV other will need cytology triage to determined appropriate next step. The risk with HPV other and any high-grade cytology meets the threshold for referrals to colposcopy (see Table 1). Australia has recently reported on the real-world experience with this triage strategy; cases with HR–HPV not 16/18 and negative or low-grade cytology (ASCUS/LSIL) at baseline and 12 months had a 3.4% risk of CIN3+ but account for over 60% of colposcopy referrals. This has led to a change in the algorithm with referrals to colposcopy only occurring if the cytology is high grade or HR–HPV 16/18 at a 12-month interval or persists over a 2-year period [20].

### 3.1.4. Triage by p16 Testing, E6/E7 mRNA and DNA Methylation

p16 is a cellular protein that is upregulated in transforming HPV infections, the test is done on cytology (or histology) slides and requires subjective assessment. This has been combined with Ki-67 staining. With a cellular proliferation marker, a positive test has at least one cell stained for both. This test is highly reproducible and, in the future, may be automated [18]. This is more specific than cytology as a triage test. Other markers of cellular proliferation include E6/E7 mRNA testing. The combination has been evaluated with the HPV-positive cytology negative group within the Italian NTCC2 trial. CIN2+ lesions did not regress if tests were positive for p16/Ki67 or E6/E7 mRNA. Hence, negative tests may be used to triage away from colposcopy as these lesions will likely resolve [37].

Methylation of host genes and viruses has been identified as a potential marker of productive infection and clinically relevant cervical lesions. Increased host DNA levels of methylation are observed as cervical epithelium progresses into a transforming CIN2/3 lesion and on to cancer. Methylation assays are available commercially and can be performed on cervical tissue, LBC samples, self-collected cervico-vaginal cells, and urine. The testing can be automated and is highly reproducible. As a testing modality, it has promise as a triage tool for HR–HPV other positive, HPV 16/18 negative results and self-collected specimens [18,38].

In summary, we recommend that HPV testing with genotyping and reflex cytology is used to triage an abnormal HPV test with a threshold for referral to colposcopy when the risk of CIN3+ is over 5%. The management algorithm for clinician collected specimens is shown in Figure 1 and the indications for entry into colposcopy in Table 2.

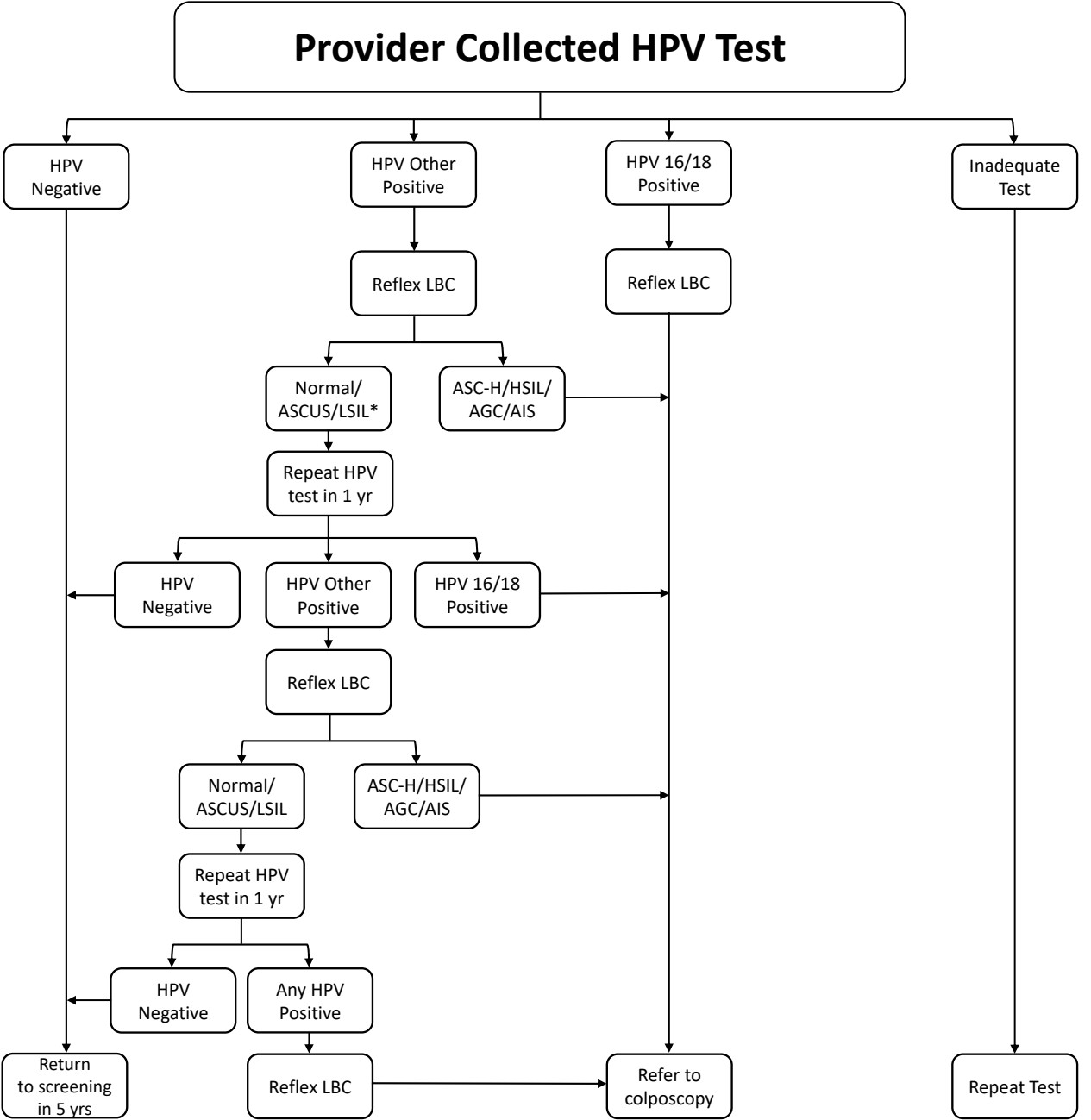

**Figure 1.** Management of Positive HPV Test Algorithm.

**Table 2.** Indications for Referral to Colposcopy.

- Patients with a positive HPV screening test should undergo HPV genotyping and reflex cytology before referral to colposcopy.
- Patients with HPV 16/18 should be referred to colposcopy.
- Patients with HPV "other" ASCUS or LSIL should have HPV testing repeated at 12 and 24 months, only referred to colposcopy if they meet other criteria or have persistent HPV "other" at 24 months.
- Patients with HPV-positive ASC–H, HSIL, AGC, AIS, or cytology suspicious for invasive cancer should be referred directly to colposcopy, regardless of HPV genotype.
- Patients who are immunocompromised with any HR–HPV should be referred to colposcopy.

### 3.2. Self-Sampling for under Screened Populations

Recommendations:

- Self-sampling may be mailed to identify non-attenders to cervical screening programs, as this has been shown in many studies to increase the uptake of screening. (strong, moderate)
- Self-sampling, coupled with face-to-face interactions, was even more effective with community health workers, nurses, or health outreach workers conducting home visits. (strong, moderate)

The availability of HPV tests has led to much interest in self-sampling with most evidence being available for under-screened populations. A systematic review by Nishimura and colleagues, consisting of 72 articles and published from 2002 to 2018, has summarized the values and preferences surrounding self-sampling and sampling settings for cervical cancer screening [39]. Overall, self-sampling was found to be more acceptable relative to clinician sampling. The attribute of completing sampling at home as opposed to attending screening at a clinic was attractive to most participants. The most studied device was the cervical swab; most women preferred it relative to other devices, such as the lavage, cervical brush, tampon, and labial padette. When compared to clinician sampling, self-sampling was viewed as simple, convenient, and more private; it induced less pain or discomfort and afforded less embarrassment or anxiety. One main concern of self-sampling was the reliability of self-sampled specimens as participants expressed greater confidence in a clinician's ability to collect specimens correctly. A unique concern highlighted in the review was the lack of face time with a clinician with self-sampling. Regardless, the authors reported that high levels of acceptability were consistent among vulnerable and under-screened populations in the review [39]. These results align with the review findings by Nelson et al., which had a pooled estimate prevalence of 59% (95% CI, 48–69%) of women preferring self-sampling [40].

Scarinci et al. published a study with the following aims: evaluate adherence to cervical cancer screening using a patient-centered approach that provided a choice of self-sampling at home for HPV testing or standard-of-care screening at the local health department ("Choice") versus only standard-of-care screening at the local health department ('SCS') among under-screened African American women [41]. "Choice" versus "SCS" was delivered by Community Health Workers (CHWs) through a door-to-door approach. A total of 335 women were enrolled in the study from 2016 to 2019. Participants in the "Choice" arm were 5.62 times more likely to adhere to cervical cancer screening compared to participants in the "SCS" arm. Women in the "Choice" arm were significantly more likely to choose (76%) and adhere to self-sampling at home with HPV testing (48% adherence) compared to standard-of-care screening at the local health department (7.5% adherence) [41].

In another study, Aboriginal women aged 25–69 years of age were recruited from eight rural and remote communities in New South Wales, Australia, to participate in HPV self-sampling via a community-based service model [42]. In total, 215 women underwent an HPV self-sampling test, and 200 evaluation surveys were completed. One-fifth of participants (*n* = 46) were never screened, and one-third (*n* = 69) were under-screened. Nine women were HPV 16/18 positive, and eight had completed all follow-ups by the conclusion of the study. A further 30 women tested positive for a high-risk type other than HPV 16/18 (HPV other), of which 14 had completed follow-up at the conclusion of the study. Satisfaction with the HPV self-sampling kit, the process of self-sampling and the service model, was high (>92% satisfied on all items). However, many women had difficulty understanding their official HPV results and needed health care provider assistance.

The option to conduct at-home self-sampling proves to be both acceptable and attractive amongst non-attenders in these studies. The reasons cited in one study for foregoing screening included feeling uncomfortable or embarrassed to be tested by a male clinician, a lack of time to attend screening, concerns of potential pain, assumptions of low risk of

HPV, and fear of a positive result [43]. Most of these concerns can be addressed with the use of mail-in self-sampling kits, thus removing barriers to access and increasing equity. However, it is important to note that screening appointments may serve as an opportunity for women to have discourse with their physician on reproductive and sexual health topics, which may be lost if all women were mailed self-sampling kits. The studies mentioned in this literature review were also primarily conducted in high-income countries. Therefore, the synthesized findings may be more generalizable to similar populations. Overall, the option to conduct an at-home self-sampling HR–HPV test proves to be both acceptable and attractive amongst non-attenders and removes barriers to access, thereby increasing equity (Table 3).

**Table 3.** Results of studies evaluating the acceptability of self-sampling HPV tests.

| Study Country | Population | N | Intervention | Results |
|---|---|---|---|---|
| Andersson et al., 2021 [44] Sweden | Women who had an HPV+ result on self-sampling and presented for diagnostic procedures | 515 (intervention) 479 (controls) | Self-sampling (kit sent by mail or opt-in online) | • The majority of participants considered self-sampling easy and reliable, and would be willing to do it again • Statistically significant differences between participants who attend screening and those who do not |
| Reiter et al., 2019 [45] United States | Women aged 30 to 65 who have not been screened for at least 3 years | 51 (intervention) 52 (controls) | Self-sampling (kit sent by mail) | • The majority of participants had a positive experience with self-sampling and would be willing to do it again • All participants preferred to do the sampling themselves at home rather than go to the doctor |
| Des Marais et al., 2018 [46] United States | Low-income women aged 30 to 64 who have not been screened in at least 4 years | 284 | Two self-samplings (one at home and the other in the clinic) | • Participants did not find self-sampling instructions hard to understand (93.6%) and most were willing to do self-collection again (96.3%) • The majority of participants had positive overall thoughts on self-collection (67.6%) |
| Maza et al., 2018 [47] El Salvador | Women aged 30 to 59 who have not been screened in at least 3 years | 1869 | Self-sampling | • Most participants agreed with statements highlighting positive aspects of self-sampling • Overall satisfaction with the experience of self-sampling was high with an average between 4.2 and 4.6 on a 5-point scale |
| Racey et al., 2016 [48] Canada (Ontario) | Women aged 30 to 70 who have not been screened for at least 30 months | 70 | Self-sampling (kit sent by mail) | • Self-collection was considered acceptable for 89.7% of participants, and 90% would be willing to do it again |

| Study Country | Population | N | Intervention | Results |
|---|---|---|---|---|
| Chou et al., 2016 [43] Taiwan | Women aged 35 to 80 who have not been screened in at least 5 years | 354 | Self-sampling | • The vast majority considered self-sampling to be simple, comfortable, and acceptable <br> • 87.2% would be willing to do it again <br> • 65.2% considered self-sampling to be a solution to their lack of attendance at screening |
| Sultana et al., 2015 [49] Australia (Victoria) | Women aged 30 to 69 years that either have never been screened or had not completed screening in the past 5 to 15 years | 1521 | Self-sampling | • Over 90% of participants found the test to be convenient, the swab easy to use, and that instructions were clear <br> • Over 80% of participants did not feel self-sampling was embarrassing or painful and felt confident that they completed sampling correctly <br> • The majority reported a preference to take their sample at home (88%) |
| Datta et al., 2020 [50] Canada (Montreal) | Women aged 21 to 65 years that either have never been screened or had not completed screening in the past 3 years | 526 | Self-sampling | • In the weighted analyses, 68% of all women surveyed and 82% of women not recently screened preferred screening by self-sampling |

Offering self-sampling among non-attenders increases screening uptake (although with varying successes of 15–43%) [51–58], when the metric used to measure the impact is the return of completed self-samples for analysis. Mailed self-sampling kits were shown to be the most effective (RR, 2.27; 95% CI, 1.89–2.71; I-squared, 99.27) and address some of the reasons reported by women that forego regular in-clinic screening, such as the lack of time or transportation, difficulties finding childcare or taking time off work, and limited access to healthcare centres [51]. Face-to-face interactions were even more effective with community health workers, nurses, or health outreach workers conducting home visits in six studies, as the likelihood of participating in self-sampling was over two times that of standard care (RR, 2.37; 95% CI, 1.12–5.03; I-squared, 99.72) [51]. Findings from this literature review have highlighted the need to ensure that only eligible women receive self-sampling kits in order to reduce wasted resources. A recent analysis of a large self-sampling trial in the USA, which included mailing HPV kits to women overdue for screening, showed that self-sampling was cost effective despite only 25% returning samples [59]. As the participation rates were highly variable across studies, it seems warranted that pilot studies be conducted to ascertain whether HPV self-sampling programs are feasible across different settings. It may also be that a multi-pronged approach that encompasses both an educational campaign to improve awareness for cervical cancer screening in concert with enabling access to self-sampling kits may be more effective in reaching a wider audience. This was evidenced in one study that reported that the combination of an educational campaign and mailed self-sampling kits was effective in engaging more women to participate in screening [60]. This literature review has found studies to show that self-sampling increases screening uptake, but even in the most successful studies, many women remained unscreened. Therefore, other options must be investigated to increase the number of women screened. Ideally,

such screening options would be effective in different settings and among diverse groups of women.

Interestingly, among the non-attenders that participated in the studies, colposcopy follow-up attendance was consistently high. This aligns with previous studies that have reported high follow-up attendance among non-attenders. Therefore, evidence suggests that adherence to follow-ups following a positive self-sampling result is generally very good.

Once again, there is a high level of evidence to support the use of self-sampling for cervical screening. Self-sampling has the distinct possibility to improve participation for that subset of the population that does not attend cervical screening. Figure 2 illustrated a suggested referral pathway; we have included the option of referral direct to colposcopy for populations that have a higher incidence of pre-cancerous changes, particularly those who have been under-screened and may be at a higher risk of pre-cancerous lesions. This will require ongoing audits and evaluation.

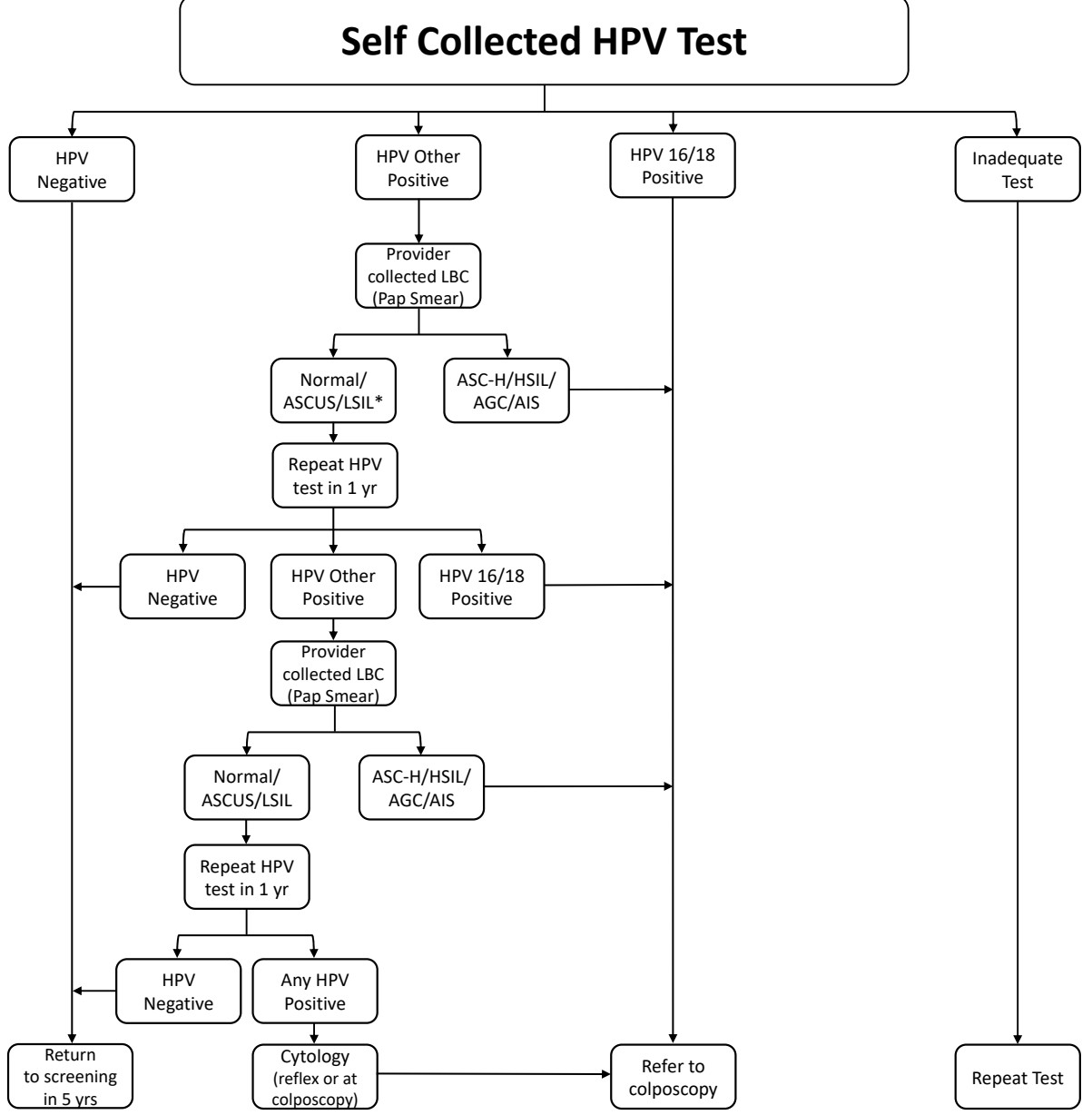

**Figure 2.** Management of a self-collected HPV test.

*3.3. Self-Sampling for the General Population*

Recommendations:

- Self-sampling could be offered to Canadian individuals with a cervix. (weak, moderate)
- Individuals should be informed that in case of a positive HPV test, they would require an appointment with a health care provider to undergo a pelvic speculum exam to obtain a Pap test or may be referred for colposcopy assessment if HR-HPV 16/18 positive. (weak, moderate)

Most of the evaluation of self-sampling to date has been in under-screened populations who may have different risk/prevalence profiles, impacting the performance of the test as a screening tool. However, there is undeniable interest for this option to be available in the general population.

A 2019 CADTH report concluded that self-sampled HPV tests had similar accuracy to clinician-collected samples to detect CIN2+ particularly when PCR-based HPV testing was conducted [61]. Updated scientific data allow for the recommendation of self-collected sampling or self-sampling to be offered to individuals with a cervix that expresses a preference for that option. In several settings, populations, and conditions, self-sampling was found to have an acceptable cost-effective profile [62].

For individuals with an adequate screening history, self-sampling provides a convenient screening method that is less intrusive, more comfortable, and less embarrassing. Self-collected sampling for HPV testing has the potential to address many of the reported challenges to screening participation, including access to a health care provider, dependence on clinic hours, and transportation. Sampling kits can be mailed to individuals who request them or could be provided in health care providers' office or other venues such as pharmacies. The exact mode of offering self-collected sampling will need to be tailored to specific jurisdictional and screening program resources. In addition, if self-collected sampling is offered as a mail-out test, programmatic involvement may be necessary, particularly for those without access to a primary health care provider.

An early meta-analysis by Petignat et al. compared self-sampling to provider sampling methods [63]. Eighteen studies (5441 participants) were included that evaluated both HR and some low-risk HPV. There was overall a high level of agreement (0.87, 95%CI 0.82 to 0.91) between self-collected and physician collected samples in terms of detection of HPV [63]. In a subsequent meta-analysis published by Arbyn, 36 studies were identified, which altogether enrolled 154,556 individuals [64]. The conclusion was that assays on self-samples were less sensitive and specific than testing on clinician-collected samples to detect CIN2+, but certain PCR-based HPV tests generally showed similar sensitivity on both self-collected samples and clinician-collected samples. The interpretation of the authors was that some PCR-based HPV tests could be considered for routine screening on self-collected samples after careful piloting assessing feasibility, logistics, population compliance, and costs. Much of the data included in the meta-analyses was collected in under-screened populations where lesions are often more prevalent and of a higher grade [64]. However, in a subsequent non-inferiority randomized trial, Polman et al. recruited women aged 29–61 years in the Netherlands to participate in a study as part of their regular screening invitation [65]. Eight thousand two hundred twelve women were randomly allocated to the self-sampling group using the Evalyn Brush and 8198 to the clinician-based sampling group. Of the samples collected, 7.4% self-collected samples and 7.2% clinician-collected samples tested positive for HPV (relative risk 1.04). The detection of CIN2+ did not differ between the two approaches [65]. The same study went on to evaluate the experience with 3835 women completing a questionnaire. Most women preferred self-sampling (76.5%) to clinicians-based sampling (11.9%) [66].

In summary, there is good evidence to demonstrate that self-sampling for HR–HPV for cervical cancer screening in the general population is not only acceptable but preferred in most cases.

Several HPV self-sampling devices and detection methods are available. Most studies report utilization of flocked swabs sent dry or in liquid medium. Generally, a self-sampling HPV test collected by vaginal swab must be low in price and offer appropriate sampling performance as well high acceptability [67]. The use of any Health Canada-approved self-sampling HPV testing device is recommended. Currently, Health Canada has approved the Rover Evalyn Brush and Copan Self Vaginal FLOQSwab collection devices, but these have not been approved in combination with the HPV test for the purposes of self-sampling [68]. A recent randomized comparison of different vaginal self-sampling devices was evaluated in 620 women referred to colposcopy for an abnormal screening test [69]. Subjects were asked to provide an initial stream urine sample collected with the Colli-Pee®device and take two vaginal self-samples, using either a dry flocked swab (DF) and a wet Dacron swab(WD), or a HerSwab(HS) and Qvintip(QT) device. Similar positivity rates and sensitivities for CIN2+ and CIN3+ were seen for DF, WD, and urine, but lower values were seen for HS and QT. Urine, a dry-flocked swab, and a wet Dacron swab all performed well and were well-received by the women, whereas the Qvintip and HerSwab devices were less satisfactory [69]. To date, there have been no safety concerns related to the use of HPV self-sampling devices reported in clinical trials.

In the general population, there is a need to have a second triage step other than HPV genotyping. In the absence of a molecular test, a cytology sample requiring a visit to a health care provider has to be obtained. This approach has been implemented in some jurisdictions [70,71]. Figure 2 shows a proposed pathway.

Australia renewed its National Screening Program in 2017 with the introduction of five yearly HPV tests. Initially, they also introduced the option of self-collection for those who had never been screened, or those who were over 4 years since the last Pap test. However, effective mid-2022 after a review of the acceptability and efficacy, a decision was made to offer all screen-eligible people the choice of self-collected or clinician-collected cervical samples. They chose to use a health care provider to mediate this [70,71]. In contrast, the BC Cancer Agency Cervix Self Screening Pilot has offered self-sampling to selected communities in the province and has implemented some ineligibility criteria, including pregnancy, history of total hysterectomy, history of AIS, history of >CIN2+ in the last 5 years, history of solid organ transplant, history of HIV+, and any active symptoms concerning for cervical cancer, such as post-coital bleeding [72]. It is recommended that, as self-sampling is implemented, cervical cancer programs monitor their performance and use their local data to inform their own practice guidelines.

### 3.4. Management of a Positive HPV Test in Immunocompromised Populations

Recommendations:

- Individuals with immunocompromised status and a positive HR–HPV test should go directly to a colposcopy, regardless of HR–HPV genotype and cytology. (conditional, low)
- Management, once in colposcopy, should follow the same guidelines as the immunocompetent population. (conditional, moderate).

Individuals who are immunocompromised have increased rates of cervical cancer, with most of the evidence being in the HIV-positive and solid organ transplant population. Individuals living with HIV have higher HPV acquisition (RR 2.64, 95% CI 2.04–3.42), lower HPV clearance (RR 0.72, 95% CI 0.62–0.84), and higher incidence of cervical cancer (RR 4.1, 95% CI 2.3–6.6) than that of the general population (Liu 2018). This also holds true for other immunosuppressed individuals [73,74]. This increased risk has led many jurisdictions to have modified screening protocols, with initial screenings to be considered between 20–24 year olds, followed by screening every 3 years until age 70, as in Australia. Others, as in the UK, do not include any modifications to the screening pathway compared to the general population [75,76]. Immunocompromised states with an increased risk of cervical dysplasia include those with: HIV, solid-organ transplants, hematopoietic stem cell transplants (especially if concomitant with graft-versus-host disease (GVHD)),

inflammatory bowel disease and rheumatoid arthritis if on an immunosuppressive agent, and systemic lupus erythematous regardless of therapy [77]. Those with inflammatory bowel disease not on immunosuppressive agents appear to have the same risk as the general population [77]. Risk-based estimates of the development of CIN3+ are currently lacking in this population. In immunocompromised patients of any age, colposcopy is recommended for all cytology results if HPV-positive [48]. Once in colposcopy, the management should be similar to the management of the non-immunocompromised population.

*3.5. Management of a Positive HPV Test in the Lesbian, Gay, Bisexual, Transgender, Queer/Questioning, Two Spirit Populations (LGBTQ2S+)*

Recommendations:

- It is recommended that primary-care providers include a process of asking patients for their identifiers, including gender at birth and their current gender to aid in the identification of patients in need of ongoing cervical screening (e.g., individuals who identify as male who have cervix). (conditional, moderate)
- HPV self-collected samples should be offered to patients who, despite a respectful and patient-centered environment, are unable to undergo provider-collected cervical samples, or simply prefer self-sampling. (conditional, moderate)
- All individuals with a cervix who have ever been sexually active should undergo routine cervical screening regardless of their gender or the gender of their sexual partners. (Strong, high)
- Referring providers and colposcopists must be respectful of gender identity and create an environment that is safe and for all individuals with a cervix regardless of gender identity or sexual orientation. (strong, moderate)

Evidence suggests that cervical cancer screening rates in the LGBTQ2S+ community are suboptimal. The implementation of primary HPV testing with an option for self-collected sampling may be more acceptable and lead to increased uptake of cervical screening.

Transgender men (i.e., men who were assigned female gender at birth) have a cervix unless they have undergone gender-affirming surgery in the form of hysterectomy. If the cervix is present, transgender men require cervical screening, as per provincial guidelines. However, studies find that rates of cervical screening in this population are less than in cisgender women. In one study, 37% of transgender men report having never received a pelvic examination or cervical Pap smear in their lifetime, compared to 5% of cisgender bisexual women reporting never having been examined. In addition, 10% reported never having had a cervical Pap smear [78]. In another study, only 58% of eligible transgender men and other non-binary people had ever undergone cervical screening [79]. In the latter study, half of patients eligible for screening felt that there was inadequate information regarding the indication for cervical screening. Barriers to screening included identifying as male, with this being indicated in the electronic health record, creating difficulty in booking appointments and obtaining results. Participation in screening also frequently involved "difficult questions", "disclosure of gender identity", and potentially negative reactions from others. Furthermore, most patients preferred not to think about that part of their body [79]. Educational materials aimed at increasing cervical cancer prevention and cervical screening are often "women–centric". This may lead to discomfort in those who identify as androgynous, transgender, or gender-queer [80]. Moreover, some health care providers have never recommended screening and others have made their patients very uncomfortable. Mistrust or apprehension regarding interactions with health care providers prevent many transgender and non-binary patient from seeking care. Fifty-three percent of patients reported that they would prefer a self-collection approach for HPV screening if it was equally effective [79].

Unmet needs for information also affect women who have sex with women. In one study, 28% of sexual minority patients perceived that cervical cancer screening was unnecessary [81]. Compared to sexual minority patients with male partners, those with only female partners are less likely to have cervical screening [82]. In a survey of young

bisexual and lesbian women, 30% of respondents had not had a Pap smear in the preceding 3 years [83]. Compared to women who have sex exclusively with men, women who have sex exclusively with women have an odds ratio of 0.10 for ever having attended cervical screening [84]. Such patients may not present as regularly for care as they may not require the need for contraception, which decreases the opportunity for discussions about cervical screening [85]. While some studies indicate a lower risk of HPV in women who identify as lesbian, this risk is influenced by the number of sexual partners, marital status, and age, and the risk is not zero [86]. In a study of 218,674 participants in the UK, the odds of CIN3+ in women who have sex exclusively with women were 1.91 compared to women who have sex exclusively with men [84]. Conversely, in the Nurse's Health Study, lesbian women had a lower odds of a positive HPV swab or abnormal cervical smear compared to heterosexual women with no same-sex partners [87]. The increased risk of CIN3+ in the former study is likely attributable to underscreening. In addition, fewer sexual minority patients undergo HPV vaccination [82,88,89]. Reported rates of smoking are also higher in the LGBTQ2S+ community, which is a known risk factor for the progression of HPV-related infections to cervical cancer [90].

Many transgender patients find the experience of cervical screening to be highly distressing. They feel vulnerable having to remove their clothes, especially if there is more than one provider in the room. While some patients prefer having a chaperone in the room, others might find this upsetting [91,92]. Speculum examinations may be painful in this patient population due to atrophy, and they may result in gender dysphoria, with the distress experienced from a mismatch between their sex assigned at birth (female) and their gender identity (male) [80]. To facilitate cervical screening, the health care community must familiarize itself with patient preferences and perform examinations that maximize patient autonomy and minimize gender dysphoria, in addition to expanding screening options that are more acceptable, such as self-swabs for high-risk HPV [93]. Many patients express a preference for HPV self-sampling, as it triggers less gender dysphoria and re-establishes control [91,92]. A positive HPV test may then provide sufficient reason to proceed with a speculum examination of the cervix [91].

The use of HPV testing as a primary screen for transgender patients may have value in patient preference and increase diagnostic value since atrophy and the use of androgens can falsely increase the findings of inadequate samples or high-grade dysplasia from cervical Pap testing [94–96]. Inadequate testing prevalence is 8.3 times higher in transgender patients, in addition to a higher likelihood for multiple inadequate tests and longer follow-up delays [80].

The performance characteristics of self-collected HPV cervical swabs versus provider-collected cervical swabs were assessed in a study of 131 transgender male patients who underwent both tests, randomized to which test was performed first. Compared to provider-collected cervical swabs as the gold standard, self-collected swabs demonstrated a sensitivity of 71% and a specificity of 98.2%. Over 90% of the study participants preferred the self-collected sample [97].

Many health care professionals have acknowledged a lack of education and some discomfort in providing primary care to patients in the LGBTQ2S+ community. For instance, 80% of health care providers surveyed received no instruction in the care of transgender patients, and only one-third of providers indicated they were comfortable caring for transgender patients [80]. Providers acknowledge that there is a lack of information, specifically clinical practice guidelines, and research to inform evidence and best practices [80]. Unsatisfactory provider–patient relationships are cited as a primary barrier to cervical cancer screening [91]. Transgender patients hope to see enhanced "cultural competence" in health care providers, with the use of appropriate pronouns and comfort with "bodies that diverged from binary male-female representations customarily presented in medical training" [91]. Some research calls for transgender-focused guidelines to address the concerns and risks specific to this population [98].

It is important to identify all patients who have a cervix, including transgender men and non-binary people (assigned female at birth). In one survey of transgender patients, only 8% had undergone hysterectomy. Meanwhile, of the remaining 92% of patients, only 27% reported a pap smear in the past year compared to 43% of cisgender patients [80]. Based on the 2019 UK Guideline for consultations requiring sexual history taking, a sexual history must include whether patients are participating in the NHS Cervical Screening Program, when their last test was taken, the result, and if they have received treatment before. This presents an opportunity for health promotion, including smoking cessation and HPV vaccination [99]. Clinic intake forms require space for transgender designation and clinic spaces should be created that are welcoming, with posters and pamphlets that are inclusive. Cervical screening programs that identify target populations using legal gender status will miss those who have legally changed their gender status [100]. Thus, the role of the primary care provider in identifying those eligible for cervical screening and creating a safe environment is of critical importance. Table 4 offers some practical guidance on how to formulate patient-centered approaches to LGBTQ2S+ patients. Box 1 provides some key definitions.

**Table 4.** Patient-Centered Approach to Care of LGBTQ2S+ Patients [92,101].

| | |
|---|---|
| (1) | Asking patients what terminology they prefer to use to describe their body parts and provide them with a safe word that would allow them to stop the procedure if needed. |
| (2) | Asking patients whether they would like to be talked through the procedure or not. |
| (3) | Giving the patient control over how the screening is performed (e.g., some patients might prefer to insert the speculum themselves). |
| (4) | Asking patients whether they prefer the procedure with or without a chaperone, since having others watching can be distressing. |

*3.6. Special Consideration and Management of a Positive HPV Test in First Nations, Inuit and Métis*

Recommendations:

- It is recommended that primary-care providers and colposcopists increase their understanding and knowledge of their local First Nations, Inuit, and/or Métis communities to work towards building relationships, knowledge translation, and understanding. (strong, moderate)
- It is recommended that colposcopy providers increase cultural safety and trauma awareness training to all persons working in the facility. (strong, moderate)
- It is recommended that colposcopists champion alternate approaches, including, but not limited to, the creation of a space and workflow that avoid re-traumatization as well as recognize local Indigenous cultures and land, addresses fears of abuse and coercion honestly, supports the patient both in the appointment and at home, and offers Cultural Support and advocacy as needed. (strong, moderate)

A study conducted in 2017 by the First Nations Health Authority (FNHA) and BC Cancer Agency found that the incidence of cervical cancer is significantly higher amongst First Nations women than non-First Nations women [102]. This study also found that First Nations people in British Columbia experience poorer overall rates of cancer survival compared to non-First Nations people. According to population data from British Columbia, cervical cancer is the fourth-most diagnosed cancer in First Nations women, compared to the seventh-most diagnosed cancer in non-First Nations women [103,104].

Higher rates of cervical cancer amongst First Nations individuals reflect the barriers they face to access the necessary preventative screenings and cancer treatment throughout Canada. Barriers may include access due to geographically available screening sites, or a lack of culturally safe cervical screening services.

Cervical cancer screening and treatment have the potential to be retraumatizing for First Nations, Inuit, and Metis peoples and communities who have experienced and/or

who may have painful memories, deep-rooted fear, and distrust of medical personnel (particularly those wearing PPE) associated with historical and personal past experiences: residential schools, Indian hospitals, provincial sterilization acts/legislation, medical experimentation, coerced temporary or permanent sterilization, sexual abuse, physical abuse, a devaluation of long-held traditional and holistic medical beliefs that further widened the gap between individuals eligible for cervical screening and their providers. Embedding cultural safety and humility into all health care services and planning is vital in mitigating these negative impacts and creating a safe health care environment where First Nations, Inuit, and Metis are eligible for cervical screening, and their families feel respected.

FNHA conducted a self-collection and acceptability pilot project for cervix screening in British Columbia. Prior to embarking on this project, Pap screening and follow-up were already in place and linked to the provincial data registry. The project showed that HPV self-collection swabs were both a feasible and preferred option for First Nations individuals, and that the utilization of self-collection swabs will support greater rates of cancer screening for First Nations individuals. Self-collected swabs were especially beneficial for trauma survivors who risk re-traumatization during a pelvic exam, particularly, with a provider unknown to them (unpublished). Of note, while First Nations community members were eager to take part in self-screening, they expressed a high level of discomfort when they were asked to attend a colposcopy appointment for follow-up after identification of an abnormal result. In more than one instance, there was refusal. Trials in Ontario and Quebec have shown similar findings regarding preference for self-screening [105–107]. Independent of the adoption of self-collection for HPV-based screening, health partners must address the fear felt by First Nations individuals and communities regarding the follow-up after a positive HPV test when recommending self-collection as an option for cervical screening. It is recommended that hospitals, clinics, and colposcopists continue working towards embedding cultural safety and humility into their services.

The proven systemic anti-Indigenous racism in the health care system, along with the ongoing impact of colonialism, has presented First Nations, Inuit, and Métis community members with many reasons to feel fear or discomfort when asked to attend a colposcopy appointment. Clients may not feel comfortable with a second or unknown provider that they have not met in advance, particularly if they have heard negative stories from friends or families about colposcopy offices being unsafe. Many First Nations individuals may not trust gynecology appointments due to fear of sterilization and the risk of being re-traumatized by their past experiences. Until 1973, BC had a law requiring the forced or coerced sterilization of certain people, of which Indigenous people were disproportionately targeted. Even after the repeal of this provincial law, Indian hospitals continued to sterilize Indigenous women without their consent. Canada's Standing Senate Committee on Human Rights released a report in June 2021 on the forced and coerced sterilizations of persons in Canada, in which they acknowledge that this practice is "not confined to the past but clearly is continuing today" and that "[its] prevalence is underreported and underestimated" [108]. Clinicians participating in cervical screening and colposcopy are encouraged to familiarize themselves with the Truth and Reconciliation Commission of Canada: Calls to Action [109]. Additional resources for clinicians are provided in Table 5. In addition, advice to clinicians and practical recommendations to improve the clinical encounter with First Nations, Inuit, and Métis are provided in Box 2 and Box 3, respectively.

*3.7. Management of a Positive HPV Test in Remote Areas, Immigrants and Newcomers to Canada*

Recommendations:

- For individuals living in rural and remote areas of Canada, HPV self-sampling techniques facilitated by mail programs or obtained from local health facilities should be strongly considered to overcome geographical barriers to cervix screening. Care pathways to obtain pap specimens and colposcopy should be in place. (conditional, moderate)
- HPV self-sampling should be considered among immigrants and newcomers to Canadian populations as an acceptable alternative to provider-collected cervical screen-

ing where cultural barriers may inhibit provider-based screening uptake. (conditional, moderate)

**Table 5.** Useful Resources to Inform Care of First Nations, Métis and Inuit Patient Populations.

BC Association of Aboriginal Friendship Centres (n.d.). *Doulas for Aboriginal Families Grant Program.* https://bcaafc.com/dafgp/

Barney, Lucy (2020). *Honouring Resilience: Providing Culturally Safe, Trauma-Informed Care for Indigenous Women and Families and COVID-19.* Available upon request.

First Nations Health Authority (2016). *FNHA's Policy Statement on Cultural Safety and Humility.* http://www.fnha.ca/Documents/FNHA-Policy-Statement-Cultural-Safety-and-Humility.pdf.

First Nations Health Authority (n.d). *FNHA's Policy on Mental Health & Wellness.* https://www.fnha.ca/WellnessSite/WellnessDocuments/FNHA-Policy-on-Mental-Health-and-Wellness.pdf.

Hope for Wellness Helpline. (n.d.). https://www.hopeforwellness.ca/.

Indigenous Corporate Training. (2020). *Indigenous Peoples and COVID-19.* https://www.ictinc.ca/blog/indigenous-peoples-and-covid-19.

National Centre for Truth and Reconciliation
Report on Truth and Reconciliation Commission of Canada: Calls to Action
https://nctr.ca/records/reports/

**Box 1.** Key Definitions & Acronyms.

**DEFINITIONS**
**Cultural humility** is a process of self-reflection to understand personal and systemic biases and to develop and maintain respectful processes and relationships based on mutual trust. Cultural humility involves humbly acknowledging oneself as a learner when it comes to understanding another's experience. (Ref: First Nations Health Authority).
**Cultural safety** is an outcome based on respectful engagement that recognizes and strives to address power imbalances inherent in the healthcare system. It results in an environment free of racism and discrimination, where people feel safe when receiving health care. (Ref: First Nations Health Authority).
**Cultural Competence** is defined as a set of values, behaviors, attitudes, and practices within a system, organization, program, or among individuals, which enables them to work effectively cross-culturally. Further, it refers to the ability to honor and respect the beliefs, language, interpersonal styles, and behaviors of individuals and families receiving services, as well as staff who are providing such services.
**Trauma-Informed Care (TIC)** is about recognizing the link between trauma and mental illness, substance use, barriers to access, physical ailments, and more. TIC involves making sure that people feel safe and are not re-traumatized by their care.
**Sex** is the classification of a person as male or female and is assigned at birth, usually based on external anatomy. It is a combination of bodily characteristics including chromosomes, hormones, internal and external reproductive organs, and secondary sex characteristics.
**Gender Identity** is a person's internal, deeply held sense of their gender. It is most commonly boy/man or girl/woman. However, some individuals are non-binary and/or genderqueer.
**Gender Expression** is the external manifestation of gender. It is expressed through name, pronouns, clothing, haircut, behaviour, voice, and/or body characteristics.
**Cisgender** is used to describe individuals who are not transgender. From Latin, cis on the same side.
**Transgender** is people whose gender identity differs from the sex they were assigned at birth.
**Gender dysphoria** is a sense of unease that a person may have because of a mismatch between their biological sex and their gender identity.
**Sexual Orientation** is a person's enduring physical, romantic, and/or emotional attraction to another person. Commonly "straight", "lesbian", "gay", "bisexual", or "queer."
**Sexual minority** is a group whose sexual identity, orientation, or practices differ from the majority of the surrounding society. Primarily used to refer to lesbian, gay, bisexual, or non-heterosexual individuals, it can also refer to transgender, non-binary, or intersex individuals.

**Box 2.** Advice for Providers When Caring for First Nation, Inuit, and Métis.

---

**Practice Cultural Humility:**
- Practice humility; know your own beliefs and honour the beliefs and practices of your clients. Treat others how you would like to be treated.
- Reflect on your own assumptions and positions of power within the health care system.
- Do not offer opinion, but only the best medical evidence. Ensure informed choice is maintained free of bias and coercion. Consult and share unbiased evidence-based resources, i.e., 1800 Sex Sense, Options for Sexual Health, SOGC.
- Listen to what the client wants, fears, and is worrying about.

**Acknowledge, Respect Culture and History:**
- Recognize that First Nations, Inuit, and Métis communities have strategies of caring for individuals from preconception to elderhood that have been passed down orally through the generations.
- Seek to draw from oral tradition by using stories to demonstrate cultural practices, beliefs, and values as evidence of healthy and protective ways of being. (Ref: Smylie).
- Learn and incorporate culture, ceremony, and traditions into care. Ensuring this education and work is done in collaboration with the community you serve.

**Facilitate Whole Person Care:**
- Work to facilitate timely access to appropriate supports for mental health and wellness, including cultural and/or traditional supports at home when possible and consider virtual care when not possible.
- Facilitate support services.
- Talk about ways to support cultural connections.
- Maintain awareness and compassion regarding possible trauma/intergenerational trauma.
- Be flexible to the needs of the community.

**Reduce Barriers for Rural and Remote communities:**
- Ensure guidelines are culturally safe, applicable to rural and remote locations, clearly communicated, and include flow charts of actions.
- Start conversations with families early to ensure there are contingency plans in place to provide a continuum of care/support for mothers and infants particularly if travelling for care.
- Be aware of the complexities of leaving/returning to communities.

---

**Box 3.** Practical Recommendations to Improving Clinical Encounter with Patients of First Nations, Metis, and Inuit background.

---

- Do not have the patient undress before seeing them.
- Build trust by calling the patient by name or meeting them in advance of appointment, allowing time to answer questions, and explain specific details of the procedure. Ideally, the person calling the patient is the person performing the examination and/or is going to be in the room during the examination (i.e., nursing).
- Consider utilization of an Indigenous liaison to be involved in communication and care.
- Allow a support person to attend the appointment.
- Acknowledge and respect concerns the patient may have regarding coerced sterilization and having an honest conversation about fertility risks.
- Provide clear next steps following the assessment/procedure ensuring community resources, and ensure supports are in place at home and known to the patient.
- Create a culturally safe and welcoming environment by using artwork by local First Nations artists to represent the community throughout the office.
- Post land acknowledgements in the office/clinic.
- Ensure all members of the team are introduced and roles defined, and the purpose of each room/piece of equipment is explained.

---

Cervical screening in immigrant populations has not been well-studied. While there are several studies evaluating cervical screening in different ethnic populations, these studies typically include a single ethnic group in a single geographic area, limiting the ability to generalize to the Canadian immigrant and newcomer populations.

In general, immigrant populations are more likely to have never been screened compared to their non-immigrant populations [50,110]. Very few studies have controlled for education, languages spoken, the country of origin, or hysterectomy status. Barriers that have been attributed to lower rates of screening uptake include insufficient knowledge about cervical screening, lower levels of English literacy, cultural differences, fear and/or embarrassment, and lack of access to female health care providers [111].

Several studies have examined the role for HPV self-sampling among immigrant populations [112–120]. Self-sampling is an acceptable alternative to clinician-collected cervical screening and may help overcome access barriers and increase participation in cervical cancer screening programs for under-screened immigrant women [112,114,116]. While self-sampling was generally perceived as favorable, some concerns were raised by study participants about whether or not they had performed the self-sampling test correctly and test accuracy, highlighting the need for education and support during the implementation of self-sampling [113,117,119,120]. The use of community health care workers was cited as a factor that increased participation [114]. In addition, for patients living in rural and remote communities, self-sampling for cervical screening was highly accepted [48]. Improving access to primary care health care providers and culturally appropriate educational material and resources should also be prioritized. At present, a pilot study for cervical screening by mail led by the BC Cancer Agency in British Columbia, Canada is already in place to help increase participation [72]. A study in Ontario is also underway, looking specifically at the acceptability and uptake of HPV self-sampling in under-screened or never-screened participants from South Asian, West Asian, Middle Eastern, and North African countries [121].

### 3.8. Management of a Positive HPV Test Following Hysterectomy

Recommendation:

- HPV vault testing is not recommended for individuals who have undergone hysterectomy for benign diseases with no prior history of abnormal pap smears. (strong, moderate)
- Patients with LSIL on hysterectomy specimen should have an HPV test at 6–12 months; if negative, they require no further follow-up. (conditional, moderate)
- Patients with a previous history of treated HSIL (CIN2/3) who had a negative HPV test thereafter and have a subsequent hysterectomy for benign indications and have no cervical pathology do not need any follow-up. (strong, moderate)
- Patients with a previous history of treated HSIL (CIN2/3) who have had no HPV-based test following treatment, have a subsequent hysterectomy for benign indications, and have no cervical pathology should have an HPV test at 12 months; if negative, they require no further follow-up. (conditional, low)
- Patients who have a hysterectomy for HSIL (CIN2/3) and have residual cervical pathology (LSIL/HSIL) should have an HPV test at 12 months; if negative, they require no further follow-up. (conditional, low)
- Patients who undergo a hysterectomy for AIS should have three consecutive annual HPV tests, followed by HPV testing every 3 years. (strong, low)
- Patients who have a history of AIS and have been discharged from colposcopy and undergo a hysterectomy for another reason should have HPV testing every 3 years. (conditional, low)
- Patients who undergo HPV testing post-hysterectomy should have reflex cytology if positive HR–HPV is detected. Referrals to colposcopy should be made for results with HR–HPV 16/18 and any cytology showing HSIL/ASC-H. (strong, moderate)
- Patients with cervical carcinoma on hysterectomy specimens are not covered by this guideline and should be followed according to gynecologic oncologist recommendation. (conditional, moderate)

Vaginal screening is not recommended for individuals who have undergone hysterectomy for benign disease with no prior history of abnormal pap smears. The risk of developing vaginal HSIL post-hysterectomy for benign indications is extremely low,

approximately 0.1–0.3%. Hence, screening for dysplasia in these patients is not recommended [122–124].

However, Vaginal HSIL can be seen in 7% of patients after hysterectomy for HSIL with most of the dysplasia being detected in the first 2–4 years post-hysterectomy. For this reason, these patients warrant some form of surveillance [123,125]. There is very little high-level evidence to advise policy with existing cytology-based recommendations being made for long-term regular surveillance. Cao, in a recent review from China, evaluated the use of HPV testing in 8581 women post-hysterectomy, 834 of whom had CIN prior to the hysterectomy. The rate of VAIN in those with a history of CIN was 7.3%, and in those that tested positive for HPV 16, there was an incidence rate of 50% of VAIN. As a result of their findings, for individuals with a history of CIN prior to hysterectomy, they recommended annual co-testing with LBC and HPV [123]. From their data, in those who had hysterectomy for CIN who subsequently had a negative HPV test, the incidence rate of VAIN was 0.7%. Furthermore, given that a negative HPV test is used to discharge individuals treated for CIN2+ or AIS from a colposcopy treated with an excisional procedure [35,126,127], it seems reasonable to extend this approach to CIN or AIS treated with a hysterectomy. Therefore, in those who had a history of CIN or CIN2+ either in the specimen or as an indication for hysterectomy, a negative HPV test should allow the screening to cease. The timing at which to perform this test is not well described in the literature. In the Cao study, the interval time to VAIN was 12.8 months. Therefore, 12 months is a reasonable time interval [123].

There is no data to guide a similar approach to those treated for AIS with a hysterectomy. The consensus is to recommend three consecutive annual HPV tests followed by HPV testing every 3 years [35,128].

Those who have vaginal testing with a HR–HPV positive should have reflex cytology. Those with HR HPV 16/18 and/or reflex cytology with HSIL or ASC-H cytology should trigger an immediate referral for vaginal colposcopy [35,123,124,129]. In individuals treated for cervical cancer with hysterectomy who received adjuvant radiation, vaginal cytology should not be performed as radiation will induce changes to the tissue that make cytology unreliable [130]. HPV testing in this specific population has not yet been established.

The working group notes that the existing literature is limited. More data is needed in this patient population to understand how best to evaluate patients post-hysterectomy. Guidelines should be updated with new data when available.

### 3.9. Screening and Management of a Positive HPV Test according to Vaccination Status

Recommendation:

- Screening and colposcopy algorithms should be the same, irrespective of HPV vaccination status. (conditional, low)

HPV vaccination was introduced in many Canadian provinces from 2008–2009. Donker et al. from British Columbia recently evaluated the effect of vaccination on CIN2+ lesions. They demonstrated a 62% (95% CI 54–68%) and 65% (58–71) reduction in CIN2 and CIN3, respectively, in the vaccinated population [131]. The UK has noted similar reductions of 97% (96–98%) for CIN3 and 87% (72–94%) for cervical cancer in vaccinated women [132]. These results make it plausible that, in the future, as vaccinated cohorts make up a larger proportion of the population targeted for screening, it will be possible to adapt screening guidelines and also decrease the intensity of screening [133]. The adverse effects of screening, with potential overtreatment of regressive CIN 2+ lesions, does have to be balanced, as incidence rates of CIN decrease in vaccinated cohorts [134]. While very exciting and promising, changes to screening in the vaccinated cohort would be premature. One of the biggest barriers to the implementation of changes to screening based on vaccination status is the inability to easily access individuals' vaccination information, either through a registry or from the individual. Furthermore, most individuals eligible for cervical screening across Canada have not been vaccinated; those who initially received the vaccine with the introduction of school-based vaccination programs when they were adolescents would now be approaching 30 years old.

Grimes et al. performed a decision analysis model to evaluate different cervical cancer screening modalities and vaccination rates on CIN2+ detection [135]. LBC-based approaches had 10 times the rate of excess colposcopy (false positive) when compared to an HPV testing and triage approach. As population vaccination rates rise to 80%, an HPV-based approach resulted in 50% less excess colposcopies compared to LBC [135]. Modelling studies have also been performed to explore the most cost-effective scenarios for vaccinated cohorts in the future. Landy and colleagues used a simulation model to conclude that the most cost-effective scenario for those vaccinated against HPV 16 and HPV 18 includes three lifetime cervical screening tests. If vaccinated against HPV 16/18/31/33/45/52/58, two lifetime screens are the most cost-effective strategy [136]. Other modeling studies similarly showed that less frequent screenings would be needed in HPV-vaccinated individuals [137,138]. Pederson et al.'s model showed screening could be delayed until 30 years old and require one-to-three screens in a lifetime, 15–20 years apart. With a later start age and less frequent screening, this has the potential not just to reduce costs, but also to decrease the harms from colposcopy [135,137,138]. However, it is important to note that these are currently only modeling studies, and that clinical data is lacking.

Given the paucity of clinical data and the lack of rigorous vaccination registries, it is recommended at this time that individuals who have received HPV vaccination should adhere to cervical cancer screening protocols that are being used for non-vaccinated individuals.

*3.10. Wait Times for Referral to Colposcopy*

Recommendations:

- Individuals with HR HPV16/18 with any cytology results should be seen within 6 weeks of referral (conditional, low)
- Individuals with an HR–HPV "other" positive test and HSIL/ASC-H/AGC should be seen in colposcopy within 6 weeks of referral. (conditional, low).
- Individuals with an HR–HPV "other" positive test meeting criteria for referral should be seen in colposcopy within 12 weeks of referral. (conditional, low).
- Individuals with an HR–HPV positive test and cytology that is suggested of carcinoma should be seen in colposcopy as soon as possible, ideally within 2 weeks of referral. (conditional, low).

Abnormal cervical screening tests and referral to colposcopy lead to significant anxiety for individuals [139]. Establishing guidelines for referral are important for establishing standards of care. Previously, the SOGC Joint Practice guidelines outlined wait times for referral to colposcopy [140]. In cases where cervical cancer is suspected, we recommend a prompt assessment in colposcopy. While there is no clear evidence of exact timelines, our working group recommends assessment within 2 weeks. Similarly, there is no clear evidence on optimal timing to referral. However, given the higher immediate risk for CIN3+ with a HPV 16 and 18 and HPV "other" HSIL/AGC/ASC-H, we recommend an assessment in colposcopy within 6 weeks. HPV "other" that is persistent over 24 months meeting criteria for referral to colposcopy should be seen within 12 weeks. This guidance is in alignment with expert opinions from other jurisdictions [127,141]. Furthermore, it is recommended that cervical screening programs established quality indicators to monitor the performance and implementation of their programs [142].

**4. Conclusions**

The recommendations in this manuscript are meant to supplement jurisdictional guidelines and not meant to replace cervical screening programs that are in place at a provincial or territorial level. We strongly support centralized programmatic-based management using consensus-based algorithms. Centralized registries to access results of previous cervical screening test are also critical for clinicians to manage patients appropriately and adhere to local guidelines. As part of the joint commission to create these guidelines, a plan is in place to implement web-based tools and applications to facilitate triaging of

abnormal test results for clinicians and when to refer to colposcopy. In future, further HPV genotyping and molecular tests may help to better triage a positive HPV test to reduce unnecessary referrals to colposcopy. A highly vaccinated population will also require less frequent cervical screening.

**Supplementary Materials:** The following supporting information can be downloaded at: https://www.mdpi.com/article/10.3390/curroncol30060425/s1, Supplementary Table S1: Literature search strategy, Search for existing clinical practice guidelines related to HPV screening; Supplementary Table S2: Literature Search Strategy—HPV testing for cervical screening in specific contexts and populations; Supplementary Table S3: Literature Search Strategy—How to manage a positive HPV test after a hysterectomy; Supplementary Table S4: Grade of Recommendations and Evaluation of Quality.

**Author Contributions:** Conceptualization, J.B., T.Z. and M.-H.M.; writing—original draft preparation, T.Z., M.-H.M., C.B., S.S., L.E., C.A., S.K., E.D., U.M., N.T., D.F. and J.B.; writing—review and editing, T.Z., M.-H.M., C.B., S.S., L.E., C.A., S.K., E.D., U.M., N.T., D.F. and J.B.; funding acquisition, J.B. All authors have read and agreed to the published version of the manuscript.

**Funding:** Production of these guidelines has been made possible through collaboration and financial support from the Canadian Partnership Against Cancer Corporation and Health Canada. The views expressed herein do not necessarily represent the views of Health Canada or the Canadian Partnership Against Cancer.

**Institutional Review Board Statement:** Not applicable.

**Acknowledgments:** This guideline was developed by the Society of Gynecologic Oncology of Canada (GOC) and the Society of Canadian Colposcopists (SCC). We acknowledge members of the Pan-Canadian Cervical Screening Network who provided thoughtful feedback to the development of the draft, as well as Jesse Ehrlick, Project Manager, Precare and Carine Trazo, Managing Director, GOC. The team wishes to acknowledge Leah Boulos, Senior Evidence Synthesis Consultant, and Kristy Hancock, Evidence Synthesis Coordinator, at the Maritime SPOR SUPPORT Unit (MSSU) for developing and executing the search strategy and assistance in writing of the methods. The team would also like to acknowledged Robin Johnson for assistance in editing the manuscript.

**Conflicts of Interest:** T.Z. has received honoraria from GSK and Merck Canada and has participated in advisory board for GSK. J.B. has received honoraria from GSK and Merck Canada and research support from the Canadian Cancer Society. The remaining authors declare no conflict of interest. The funders had no role in the design of the study; in the collection, analyses, or interpretation of data; in the writing of the manuscript; or in the decision to publish the results.

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
