# Peer review of "Canadian Guideline on the Management of a Positive Human Papillomavirus Test and Guidance for Specific Populations"

_curroncol, doi:10.3390/curroncol30060425_

Round 1

Reviewer 1 Report

The manuscript reports clinical practice recommendations  on the management of a positive HPV test in the context of cervical cancer screening, to supplement the Canadian guidelines. It also includes very detailed indications and recommendations to guide the management of specific populations (i.e., underserved and marginalized groups). The contents are well documented and well presented, can usefully serve to implement and personalise the management protocols (with value for both the population and the healthcare professionals), and fully comply with the intended aim.

MINOR COMMENTS:

-TITLE: in my opinion, the aim "to supplement jurisdictional guidelines" should be made evident also in the title.

-METHODS: since the paper will have a larger than Canada audience, a sentence on how the cervical screening program is operated (age range, type of primary test, interval) in Canada should be added.

-Paragraph 3.8 (line 687): please check the title because it is not clear (is the term "apposite" correct?).

-Lines 701-703: the performance of only one HPV test at 12 months without further follow-up in women who underwent hysterectomy for HSIL (CIN2/3) and have residual cervical pathology (LSIL/HSIL) seems inadequate, and further follow-up should be provided.

-Paragraph 3.9, lines 753-755: studies and evaluations done in the last years have already reported some indications for modifying the screening protocol for women fully vaccinated before 15 years of age, such as screening initiation not earlier than 30 years by HPV testing (see: Prev Med 2017;21-30). As also noted by the authors in lines 768-769, protocols modifications need access to vaccination registry to be integrated with screening database to perform a taylored strategy. Since in Canada HPV vaccination was introduced in 2008/2009, the women born in 1997/1998 will reach 25 years in 2022/2023, changes in the protocol need time for implementation. To help preparing all the stakeholders to change, this should be made evident in the recommendation. 

Some minor spelling mistakes are present.

The citation in the text of a couple of references is displaced (ref.13) or missing (ref.45).

Author Response

Thank you for the comprehensive review of our paper. We will answer your comments on order:

-TITLE: in my opinion, the aim "to supplement jurisdictional guidelines" should be made evident also in the title.

WE have considered how to address this and the aim to be  supplementary to jurisdictional guidelines is covered in both the abstract and introduction in the 

-METHODS: since the paper will have a larger than Canada audience, a sentence on how the cervical screening program is operated (age range, type of primary test, interval) in Canada should be added.

We were not commissioned to look at this and certainly we currently do not know how each province may introduce HPV testing, there may be variability in age of onset, cessation and interval. We have added a sentence to cover this:

Primary HPV testing currently has not been implemented, however it is anticipated that this will happen soon. The age of initiation, cessation and interval are controlled at the provincial level and were beyond the scope of this document.

-Paragraph 3.8 (line 687): please check the title because it is not clear (is the term "apposite" correct?).

-Corrected

-Lines 701-703: the performance of only one HPV test at 12 months without further follow-up in women who underwent hysterectomy for HSIL (CIN2/3) and have residual cervical pathology (LSIL/HSIL) seems inadequate, and further follow-up should be provided.

We agree that this is a very challenging situation come up with a recommendation, that ASCCP currently state annual HPV x 3. We found the risk of VAIN from the Chinese study (Cao et al) was ultimately very low in those individuals who had a negative HPV test following hysterectomy with for CIN, the incidence rate was 0.7%. Given this, we feel that cessation of screening following a negative HPV test is safe and does not warrant ongoing vault surveillance. This also is in accordance with at least one large provincial guideline that will be released soon. For these reasons and the low rate after a negative HPV test the committee were comfortable with one negative test a t 12 months.

-Paragraph 3.9, lines 753-755: studies and evaluations done in the last years have already reported some indications for modifying the screening protocol for women fully vaccinated before 15 years of age, such as screening initiation not earlier than 30 years by HPV testing (see: Prev Med 2017;21-30). As also noted by the authors in lines 768-769, protocols modifications need access to vaccination registry to be integrated with screening database to perform a taylored strategy. Since in Canada HPV vaccination was introduced in 2008/2009, the women born in 1997/1998 will reach 25 years in 2022/2023, changes in the protocol need time for implementation. To help preparing all the stakeholders to change, this should be made evident in the recommendation. 

Thank you for this comment. We believe we have discussed the challenges with individualized screening based on vaccination status, in the absence of a universally accessible vaccine registry this would be very difficult to do. At this time there are no national screening programs that alter screening based on vaccination status. In the paper you quote the may consider starting at 30 but they also were unable to extend interval beyond 5 years. As we stated we believe that more clinical data is needed before change based on vaccination status.

Comments on the Quality of English Language

Some minor spelling mistakes are present. -->

The citation in the text of a couple of references is displaced (ref.13) or missing (ref.45).

Thank you for noting this. Ref 13 appears to be corrected

-Ref 45 was incorrectly assigned by endnote. It is now corrected and removed. 

Reviewer 2 Report

The authors present a comprehensive review of screening strategies for cervical lesions induced by persistent HPV infection. The merit of the work lies not only in its scientific rigor but also in its responsible and inclusive approach to marginalized population groups or those with limited access to medical care. The presentation is very well formulated and the results are presented and discussed in an exemplary way.

Author Response

Thank You for the review

Reviewer 3 Report

This paper provides guidelines for the management of specific patient populations among patients who test positive for HPV. Authors summarize as much evidence as possible for the handling of patient populations in special settings, such as immunocompromised, LGBTQ2S+, indigenous, and post-vaccinated patients, in addition to methods of testing and some triage methods.

This guideline was produced in cooperation with the Canadian Society of Gynecologic Oncology, and the content is not particularly problematic. The key points in each chapter are appropriately summarized and very easy to read.

Author Response

Thank you for the review